# Patients’ Perceptions and Experiences during the Pre-Admission Phase for Total Hip Replacement Surgery: A Qualitative Phenomenological Study

**DOI:** 10.3390/jcm12082754

**Published:** 2023-04-07

**Authors:** Umile Giuseppe Longo, Anna Marchetti, Alessandra Corradini, Vincenzo Candela, Giuseppe Salvatore, Maria Grazia De Marinis, Vincenzo Denaro

**Affiliations:** 1Research Unit of Orthopaedic and Trauma Surgery, Fondazione Policlinico Universitario Campus Bio-Medico, Via Alvaro del Portillo, 200, 00128 Roma, Italy; a.corradini@policlinicocampus.it (A.C.); v.candela@policlinicocampus.it (V.C.); g.salvatore@unicampus.it (G.S.); denaro.cbm@gmail.com (V.D.); 2Research Unit of Orthopaedic and Trauma Surgery, Department of Medicine and Surgery, Università Campus Bio-Medico di Roma, Via Alvaro del Portillo, 21, 00128 Roma, Italy; 3Research Unit Nursing Science, Campus Bio-Medico University, Via Alvaro del Portillo, 21, 00128 Rome, Italy; a.marchetti@unicampus.it (A.M.); m.demarinis@unicampus.it (M.G.D.M.)

**Keywords:** hip osteoarthritis, pain, quality of life, qualitative research

## Abstract

Osteoarthritis negatively impacts the patient’s quality of life. Qualitative research is an effective tool in detecting the various emotions experienced by patients with osteoarthritis. Such studies play a crucial role in promoting comprehension of the patient’s experiences of health and illness among healthcare professionals, including nurses. The purpose of this study is to examine patients’ perceptions during the pre-admission process for total hip replacement (THR). The study utilized a qualitative descriptive methodology with a phenomenological approach. A sample of patients awaiting THR consented to participate in the study and were interviewed until data saturation was achieved. The results of the phenomenological analysis resulted in the identification of three themes: 1—Surgery generates mixed feelings; 2—Pain negatively impacts daily life activities; 3—Pain requires personal strategies to be alleviated. Patients awaiting THR demonstrate frustration and anxiety. They experience intense pain during daily activities, which persists even during night rest.

## 1. Introduction

Osteoarthritis is widely recognized as the most prevalent joint disease globally. Specifically, hip osteoarthritis (HO) is particularly linked to significant societal limitations and associated impairments [1,2]. Pain is a common symptom experienced by individuals with this condition. As a result, the focus must be on the characteristics of the pain symptomatology, including its location, triggering factors or remission, intensity, and duration. Total hip replacement (THR) is considered an effective intervention for improving functionality and alleviating pain [3,4].

Surgical intervention should be considered for patients who have not experienced improvement through less invasive treatments, including physiotherapy, injections, or pain management with analgesics [4,5,6]. THR is performed more on the aging population, which has several risk factors for developing depression, such as social isolation, lower socioeconomic status, general medical conditions, uncontrolled pain, and decreased sleep quality [7,8].

In addiction to pain, disability and participation restriction are important aspects to assess in the care plan [9]. These patients experience worsening quality of life and a decreased ability to perform activities of daily living [3,10].

At present, there are limited studies that have assessed the qualitative aspects of patient experience following THR. Some studies suggest that patients report improved quality of life, reduced pain, and improved mobility after THR. However, other studies report dissatisfaction with prosthesis function, aesthetic appearance, and limited range of motion. Research is needed to fully understand the patient’s perspective on THR and to develop a standardized approach to evaluating patient satisfaction [11].

Qualitative research is probably the best way to understand patients’ needs and contexts and could improve therapeutic strategies and their assessment [9]. Knowing the patient’s emotions and feelings pre-operatively can help solve problems earlier [12].

This study aims to explore patients’ experiences during pre-admission for THR.

## 2. Materials and Methods

### 2.1. Design

Qualitative research examines, characterizes, and interprets the human experience, and interviews are predominantly utilized to generate qualitative data [13].

This study employed a qualitative research design. By utilizing qualitative research, the researcher gained an understanding of the patients’ beliefs and values and the way these beliefs and values are shaped by the context in which they are studied [14]. Interviews are a prevalent technique for data collection in nursing research [15]. These interviews were conducted using a phenomenological method, as it facilitates the understanding of the patient’s internal world, including emotions, perceptions, experiences, and visions.

The descriptions are exhaustive and expressed in a language that is accessible to those who experience the phenomenon.

Giorgi’s analytical method endeavors to uncover the essence of the phenomenon as perceived by a human, through the detection of crucial themes [16]. This method consists of five steps:Collection of verbal data;Reading of the data;Breaking of the data into parts;Organization and expression of the data from a disciplinary perspective;Synthesis or consolidation of the data for the expression of the pattern of the phenomenon [17].

### 2.2. Participants

Data were obtained through unstructured interviews. The interview scheduling was based on the pre-admission list for surgical procedures. The inclusion criteria were as follows: adult patients undergoing pre-admission for THR due to HO and having given informed consent to participate in the study. The exclusion criteria were as follows: patients who did not give consent to participate in the study and to use the data; age under 18; no knowledge of the Italian language; and cognitive deficits that do not allow them to describe their experience. All participants received clear explanations regarding the research objectives and interview methods. The participants consented to take part and were interviewed until sufficient information was obtained. The conclusion of the interviews was determined by the absence of significant information from subsequent participants.

### 2.3. Setting and Data Collection

From July 2022 to August 2022, data were gathered through qualitative interviews at the Orthopaedic and Traumatological Surgery Department of a university hospital in Rome, Italy. Participants were selected based on the pre-admission list for surgery and interviewed. The interviews, which lasted between 5 and 15 min, were digitally recorded and transcribed verbatim.

Prior to conducting the interviews, the research team discussed the central themes of the study with the participants and encouraged them to provide narrative responses to the questions. The participants then provided informed consent.

The questions were designed to allow respondents the flexibility to direct the flow of the conversation and structure it to ensure that critical topics were addressed.

The following questions were asked: Can you tell me about your health problem? What changes (physical, psychological, emotional, etc.) did it bring about in your life? What are your fears (emotions/thoughts) about the future of your illness? How do you sleep? How is the quality of sleep? Would you like to add more?

### 2.4. Data Analysis

The data analysis was performed using Giorgi’s descriptive phenomenological method. During the first step, the researcher collected verbal data through audio recording the interviews. Each day, the interviews were transcribed verbatim to ensure the accuracy of the collected data. This transcription preserved the exact wording of the participants. During the second step of the data analysis, the researcher thoroughly reviewed the transcriptions with an open mind to gain a general understanding of the information gathered. In the third step, the transcriptions were divided into smaller segments, and each segment was re-examined line by line to identify the “units of meaning,” which are distinct and meaningful units within the descriptions. In the fourth step, each identified unit of meaning was analyzed and reorganized to enhance its disciplinary significance and express it in terms relevant to the clinical context. Finally, in the last step, similar units of meaning were grouped and combined into larger subcategories to synthesize the meaning of the data. In the abstraction phase, subcategories with conceptual and semantic similarities were grouped into generic categories. These categories were intended to represent the underlying essence of the data, and the relationships among them were made explicit through the identification of themes and sub-themes.

A senior researcher experienced in qualitative research oversaw all phases of the analysis. The research team engaged in discussions to resolve any discrepancies until a consensus was achieved.

### 2.5. Trustworthiness

To ensure the validity and reliability of the results, the Lincoln and Guba criteria were adopted, which evaluate the trustworthiness of qualitative research in terms of credibility, reliability, confirmability, and transferability. To further enhance credibility, the research continued sampling and data collection until saturation was reached. Reliability was ensured using the triangulation technique, which requires the involvement of multiple researchers in the analysis process. This provides multiple perspectives and confirms the results, adding depth and breadth to the study phenomenon and leading to multiple conclusions. Confirmability was established with audit trails, in which two independent researchers reviewed the survey process to ensure that the results were supported by the data. Transferability was demonstrated through a description of the socio-demographic characteristics of the participants and a thorough and accurate description of the methodology, research process, and results.

### 2.6. Ethical Considerations

The study was conducted according to the Declaration of Helsinki and was approved by the Institutional Review Board of Campus Bio-Medico University of Rome (COSMO study, Protocol number: 78/18 OSS ComEt CBM, 16/10/18).

Eligible patients were contacted by telephone on their pre-admission day and were provided with verbal information about the purpose of the study, the mode of the interview, and the requirement for digital audio recording. Prior to the start of the interview, all participants signed an informed consent form to indicate their agreement to participate in the study. Anonymity was protected by identification codes during the transcription of the interviews and throughout the data analysis process, ensuring that no individual participant could be traced in the study report. Participants were informed that they had the option to withdraw from the study at any time.

## 3. Results

The study included a sample of 12 patients who were awaiting THR. The sample consisted of 7 females and 5 males with a mean age of 69 years (Table 1).

Three main themes are deduced from this phenomenological analysis. The first theme is “Surgery generates mixed feelings”; the second theme is “Pain negatively impacts daily life activities”; the third theme is “Pain requires personal strategies to be alleviated”.

### 3.1. First Theme: Surgery Generates Mixed Feelings

The results of the qualitative study indicated that patients undergoing THR experience a range of emotions. Firstly, there is a general fear associated with their condition. This fear is mainly due to the patient’s personal medical history and the uncertainty generated by the waiting period for the surgery. Many patients reported that they do not know when the surgery will take place or how the postoperative period and recovery process will unfold, leading to frustration and numerous thoughts:

“[B]ecause I have a job, I’m worried about the post operation: it’s long and so I’ll have to figure out how to handle it because it’s actually quite challenging, we talk about an abundant month, so this gives me a bit of tension…”(F, 73 years)

“Now I’m like this: I’m waiting, then I hope I’m okay, I think Oh, God, if it’s not okay I can’t walk anymore. I get three thousand troubles in my head…”(F, 72 years)

The results of the qualitative study indicate that patients undergoing THR exhibit a range of emotions. A general sense of fear and uncertainty prevails due to their personal medical history and the wait for the surgery. Participants expressed frustration and various thoughts about the unknown timing of the surgery and the postoperative and recovery process. Despite this, many patients also expressed optimism, confidence, and hope that the intervention will alleviate their pain and physical limitations caused by osteoarthritis, allowing them to return to their daily life activities. The surgery is perceived as the only solution for many participants who have had to abandon hobbies because of osteoarthritis:

“I hope to improve. I’m optimistic; I’ve forgotten how to walk without feeling pain, without limping, it’s something I see far. I want to be what I was…”(F, 59 years)

The feelings of discouragement and frustration are seen in many patients, mainly because of the fear of not being able to return to the workplace or to their usual activities. Many patients report that osteoarthritis and the consequent decrease in mobility have led to the abandonment of various activities they perform daily, such as work, spending time on the house, playing with grandchildren, or walking:

“I’m a bit depressed, because I volunteer, I was with my niece, I went around, Fiumicino-Rome, I took buses, I was a person who walked a lot. Then I stopped and I’m depressed…”(F, 69 years)

“I feel this fact very heavy; it depresses me, I have a little depression because I can’t do things, I can’t walk…”(F, 72 years)

Waiting for surgery also generates a feeling of nervousness and fatigue in these patients:

“I’ve been fighting with this hip for years, now I’ve really reached the limit…”(F, 67 years)

### 3.2. Second Theme: Pain Negatively Impacts Daily Life Activities

HO can result in decreased mobility and significant physical disability that affects an individual’s independence in performing daily activities such as walking, ascending stairs, bathing, and rising from a seated position:

“It changes your life, you don’t live anymore… making a move in the bed is a sacrifice, taking a shower is a sacrifice, putting on pants is a sacrifice…”(M, 56 years)

“You can’t make certain movements: my leg doesn’t allow me, even to put a sock, I need to put myself in suitable positions to do it…”(F, 63 years)

This reduced mobility is reported not only during normal daily activities but also during night rest when patients experience severe pain leading to a worsening of the quality and quantity of sleep. Night pain is, in fact, a complex problem, affecting both the beginning of sleep and the maintenance of the same.

“When I’m in the bed I can’t turn around with this leg…”(M, 70 years)

“[A]lso sleeping is a problem… making a movement in the bed is a sacrifice…”(M, 56 years)

“I sleep badly, with pain I cannot turn in bed…”(F, 69 years)

“I don’t have a good quality of sleep, I find it hard to turn around in bed…”(F, 59 years)

### 3.3. Third Theme: Pain Requires Personal Strategies to Be Alleviated

Most of the study participants reported that they had utilized various methods to alleviate their pain prior to seeking care at the orthopedic department and undergoing surgery. The initial management of coxarthrosis typically includes FANS, educational programs, exercise, weight loss (for eligible patients), corticosteroid injections, or hyaluronic acid injections, as well as other medications:

“I also did injections of hyaluronic acid, but they didn’t help so I raised my hands and thought it was time for the surgery…”(F, 59 years)

“Now I’m applying ice pack[s], because I did punctures, dicloreum, contramal, they work on me and then I took Oki but now even that so I’m trying the ice…”

We recommend an exercise program that does not involve high-impact activities. Water exercises also improve function:

“I used to ski, I couldn’t do it anymore. I couldn’t do many sports. Same issue with [t]ennis was the same, I avoided it. The only sport I could do was the pool…”(M, 71 years)

Most of the patients reported pain during the night rest, and a majority of them implement the same strategy to improve the quality of sleep:

“I must sleep with the pillow between my legs… otherwise I have to put my hand on the buttock when I lie down, so I feel better…”(M, 56 years)

“I sleep badly, with a pillow between my legs and always in the same position, I cannot turn around because I have continuous pain even at night…”(F, 67 years)

## 4. Discussion

Literature has shown that the pre-operative experiences of patients undergoing THR are commonly reported. With regard to the first theme, “Surgery generates mixed feelings,” from the literature emerges the conclusion that patients feel that a sense of uncertainty prevails about the surgery and, specifically, what treatments should have been undertaken before. The most prevalent belief among patients is that undergoing surgery was an inevitable turning point from which there was no return [18]. Another sentiment reported by patients with HO is anxiety related to the waiting period for surgery. Prolonged waiting times for THR can result in increased financial costs and decline in physical function during the waiting period [19].

Patients who do not achieve adequate pain relief and functional improvement from conservative treatments may be eligible for surgical intervention. However, treatment may not always align with guidelines, recommendations, or patients’ preferences, leading to dissatisfaction and decreased treatment adherence [20,21].

With regard to the second theme, “Pain negatively impacts daily life activities,” the literature confirms that the positive effect of THR on health-related quality of life is well established. Patients with advanced arthrosis commonly experience depressive symptoms due to their inability to carry out routine activities, resulting in a lack of physical strength, stability, and agility. They report difficulties in performing household chores and are often compelled to give up their cherished hobbies, such as walking, playing sports, or spending time with their grandchildren. This forced sedentary lifestyle contributes to depression, by definition. Significant improvement after THR was observed in the dimensions of motion, discomfort, and symptoms, as well as in the dimensions of distress, daily activities, and vitality [22].

The importance of effectively managing pain and receiving proper information and support to increase confidence in early discharge programs and postoperative management has been established [23]. Persistent pain is a complex and individualized experience that is the primary symptom of osteoarthritis and can result in decreased health quality and disability [24].

In orthopaedic patients, pain assessment is often conducted using Visual Analog Scales (VAS). However, these scales have several limitations, as they fail to take into account the intricacy of an individual’s pain experience [25].

Patients with osteoarthritis (OA) often experience exacerbated night pain, resulting in a decrease in sleep quality. This decrease in sleep quality has been shown to correlate with the radiographic severity of the disease. Pain has been established as a contributing factor to reduced sleep quality, and decreased sleep quality has been linked to heightened pain sensitivity [3]. Night pain is a significant pain experience in hip OA [25]. In this study, to assess the mentioned night pain, we used simple questions such as “How do you sleep?” or “How is the quality of your sleep?”

Night pain is a multifaceted issue that impedes both sleep onset and sleep continuation. In terms of decreased sleep quality, patients often face difficulties finding and maintaining a comfortable sleep position, leading to reductions in both sleep quality and quantity. THR has been observed to alleviate osteoarthritic hip pain in a significant number of patients [26]. The quality of sleep is primarily influenced by pain; however, to a large extent, it also depends on physical difficulties. Patients report not being able to turn over in bed due to the stiffness of the joint and the reduction in internal rotation inherent in coxarthrosis. There also seems to be a correlation between sleep quality and depression in these patients. In fact, it is known that coxarthrosis increases the risk of sleep disorders and that both pain and sleep problems can trigger functional disability and depression, which seems to play an important role in the sleep-pain link, particularly when the pain is severe [27].

With regard to the third theme, “Pain requires personal strategies to be alleviated,” from the literature emerges the conclusion that a lot of patients try to alleviate pain through a conservative method such as the use of FANS, hyaluronic acid or corticosteroid infiltrations, or physiotherapy. As reported in the literature, patients tend to exercise caution in utilizing pharmacological interventions, such as drugs or injections, to alleviate their symptoms due to the potential for adverse effects, including infections, gastrointestinal issues, hypertension, and headaches. In addition, patients express concern about the possibility of developing tolerance to the medications or having to escalate their dosage over time to maintain symptom relief. This fear of drug dependence or resistance further contributes to their reluctance to utilize pharmacological interventions [20].

Consequently, many patients do not comply with their prescribed pain medication regimen, leading to difficulties in effectively managing pain in their daily lives. This often includes a reduction in physical activity, which can result in exacerbated pain and decreased physical function [28].

However, if the pain persists for more than six months, the orthopedic surgeon recommends surgical treatment for them to reduce the symptomatology.

Individuals’ beliefs regarding chronic pain influence their attitudes and behaviors towards pain management. There is often confusion regarding the origin and variability of chronic pain, and without sufficient information and guidance from healthcare providers, individuals may lack knowledge about appropriate actions to take and instead adopt a strategy of avoiding physical activity to mitigate the risk of further harm [29].

## 5. Conclusions

This study shows how patients feel waiting for a THR surgery. They are frustrated and discouraged and experience great fear. The major problem related to these sensations is the pain that is present at any time of the day, even during night rest, and this implies a worsening of the quality of life. To manage this pain, many patients use a conservative method, which, in the long run, leads to fear of not being able to recover completely. These results help professionals better understand the patients’ experiences and to anticipate some needs: this is important for making the patient feel involved in the treatment, as well as adding content to the pre-operative educational process, which also improves post-operative outcomes.

## Figures and Tables

**Table 1 jcm-12-02754-t001:** Characteristics of participants.

Patients’ Characteristics	N (%)
Age	
<60	16.6%
60–70	41.6%
>70	41.6%
Gender	
*Male*	41.6%
*Female*	58.3%

## Data Availability

The datasets used and/or analysed during the current study are available from the corresponding author upon reasonable request.

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
