# Peer review of "Patients’ Perceptions and Experiences during the Pre-Admission Phase for Total Hip Replacement Surgery: A Qualitative Phenomenological Study"

_jcm, 2023, doi:10.3390/jcm12082754_

Round 1

Reviewer 1 Report

Qualitative analysis on preoperative perception of patients with Osteoarthritis awaiting Total Hip Replacement.

Qualitative description of study, backgrounds, methods and discussion was satisfactory. My concern only 12 patients were sampled on views prior to THR. There was no in depth questioning of patients and conclusions generated were simplistic- Ie patients had mixed feeling approaching surgery, had persistent pain issues and had anxiety awaiting surgery.

Need s more advanced qualitative analysis ( ie description of patient's functional status, quality of life (QOL) indicators should have been concluded. Needs a greater numbers of patients in descriptive study  and can then draw more extensive conclusions.

Author Response

Thank you for your comment. In qualitative research, as reported in the following article, the data sample size is small and dependent on data saturation. Sample size differs for each study. In this type of research, usually small sample size, is typically determined by the richness of the data, diversity of the participants or units, scope of the research question, data collection method, and sampling strategy employed. The decision to end data collection and determine sample size is usually made collaboratively by the research team when data saturation is achieved. Phenomenological studies, such as the present one, typically require fewer than 10 interviews. [1]

1- Moser, A. and I. Korstjens, Series: Practical guidance to qualitative research. Part 3: Sampling, data collection and analysis. Eur J Gen Pract, 2018. 24(1): p. 9-18.

Reviewer 2 Report

This manuscript aimed to examine the patients' perceptions during the pre-admission process for total hip replacement (THR). The 20 study utilized a qualitative descriptive methodology with a phenomenological approach. A sample of patients awaiting THR consented to participate in the study and were interviewed until data saturation was achieved. The results of the phenomenological analysis resulted in the identification of three themes: 1- Surgery generates mixed feelings; 2- Pain negatively impacts daily life activities; 3- Pain requires personal strategies to be alleviated.

I read the article with interest, the title reflects well the content of the study

A)   The abstract is sufficiently developed, but a few concerns are present:

Comment 1 Clear reference should be made to the characteristics of the study.

B)    In the introduction, the characteristics of the hip osteoarthritis have been accurately described, even if a little too synthetic.

Comment 2: Some references should be added regarding the diagnosis, treatment, and prognosis that can occur after this condition, for example: (Sessa G, et al (2019) " Bone mineral density as a marker of hip implant longevity: a prospective assessment of a cementless stem with dual-energy X-ray absorptiometry at twenty years").

Comment 3: " Pain is a common symptom experienced by individuals with this condition. As a result, the focus must be on the characteristics of the pain symptomatology, including its location, triggering factors or remission, intensity, and duration.”  Please, adding some bibliographic references.

Comment 4: “THR performed more on an aging population, which has several risk factors for developing depression such as social isolation, lower socioeconomic status, general medical conditions, uncontrolled pain, and decreased sleep quality.” Please, adding some bibliographic references.

C)    In materials and methods, the evaluation methods have been adequately developed.

Comment 4: It is not appropriate to specify the aim of the study in methods.

Comment 5: Furthermore, it would be more appropriate to group exclusion criteria in order to be clearer to the reader.

D)   The discussion is sufficiently developed, even if a little too synthetic.

Comment 6: It is not appropriate to specify the aim of the study in the discussion.

Comment 7: It would be appropriate to use the reference style of the magazine, [1], [1-2] or [1,2].

Comment 8: It would be worth referring to the limitations of your study.

Finally, English language editing is needed.

Nevertheless, some major changes are needed.

Author Response

This manuscript aimed to examine the patients' perceptions during the pre-admission process for total hip replacement (THR). The 20 study utilized a qualitative descriptive methodology with a phenomenological approach. A sample of patients awaiting THR consented to participate in the study and were interviewed until data saturation was achieved. The results of the phenomenological analysis resulted in the identification of three themes: 1- Surgery generates mixed feelings; 2- Pain negatively impacts daily life activities; 3- Pain requires personal strategies to be alleviated.

I read the article with interest, the title reflects well the content of the study

  1. A)   The abstract is sufficiently developed, but a few concerns are present:

Comment 1 Clear reference should be made to the characteristics of the study.

  1. B)    In the introduction, the characteristics of the hip osteoarthritis have been accurately described, even if a little too synthetic.

Comment 2: Some references should be added regarding the diagnosis, treatment, and prognosis that can occur after this condition, for example: (Sessa G, et al (2019) " Bone mineral density as a marker of hip implant longevity: a prospective assessment of a cementless stem with dual-energy X-ray absorptiometry at twenty years").

  • Thank you for the suggestion. We added some references [2-4]

Comment 3: " Pain is a common symptom experienced by individuals with this condition. As a result, the focus must be on the characteristics of the pain symptomatology, including its location, triggering factors or remission, intensity, and duration.”  Please, adding some bibliographic references.

  • Thank you for the suggestion. We added some references [5, 6]

Comment 4: “THR performed more on an aging population, which has several risk factors for developing depression such as social isolation, lower socioeconomic status, general medical conditions, uncontrolled pain, and decreased sleep quality.” Please, adding some bibliographic references.

  • Thank you for the suggestion. We added some references [7, 8]
  1. C)    In materials and methods, the evaluation methods have been adequately developed.

Comment 4: It is not appropriate to specify the aim of the study in methods.

  • Thank you for the clarification. As per your suggestion, we have removed the study's aim description from the methodology section.

Comment 5: Furthermore, it would be more appropriate to group exclusion criteria in order to be clearer to the reader.

  • Thank you for your pertinent comment. We agree with you and, based on your suggestion, have added the group exclusion criteria. The exclusion criteria were: patients who did not give consent to participate in the study and to use the data; age under 18; no knowledge of the Italian language; cognitive deficits that do not allow them to describe their experience.
  1. D)   The discussion is sufficiently developed, even if a little too synthetic.

Comment 6: It is not appropriate to specify the aim of the study in the discussion.

  • Thank you for the clarification. As per your suggestion, we have removed the study's aim description from the discussion section. Furthermore we increased the discussion.

Comment 7: It would be appropriate to use the reference style of the magazine, [1], [1-2] or [1,2].

  • Thank you for your comment. We modified the reference style of the magazine.

Comment 8: It would be worth referring to the limitations of your study.

  • Thank you for your appropriate comment. We added the limitations of the study.

Some limitations of the study may be:

Sample representativeness: the sample of participants may not be sufficiently representative of the population of interest, limiting the generalisability of the results. The study only enrolled Italian patients visiting the Orthopaedic and Traumatological Surgery Department of a University Hospital in Rome, Italy. It may be worthwhile to consider recruiting patients from other regions of Italy, who, despite sharing a homogeneous culture, may present inequalities due to different daily habits.

Data collection limitations: the collection of qualitative data often depends on the participants' responses, which may not always be accurate or complete.

Finally, English language editing is needed.

  • The manuscript has been thoroughly revised by a native English speaker. The reviewer checked the grammar, spelling, punctuation and style of the text and made corrections and suggestions where necessary. We hope that the revised manuscript is now clear, concise and consistent in scientific English.

Nevertheless, some major changes are needed.

1- Lespasio, M.J., et al., Hip Osteoarthritis: A Primer. Perm J, 2018. 22: p. 17-084.

2- Katz, J.N., K.R. Arant, and R.F. Loeser, Diagnosis and Treatment of Hip and Knee Osteoarthritis: A Review. JAMA, 2021. 325(6): p. 568-578.

3- Ferguson, R.J., et al., Hip replacement. Lancet, 2018. 392(10158): p. 1662-1671.

4- Hunter, D.J. and S. Bierma-Zeinstra, Osteoarthritis. Lancet, 2019. 393(10182): p. 1745-1759.

5- Hönle, W., et al., [Conservative and surgical therapy of hip osteoarthritis]. MMW Fortschr Med, 2007. 149(37): p. 31-3, 35.

6- Wang, S., et al., Depression and anxiety symptoms are related to pain and frailty but not cognition or delirium in older surgical patients. Brain Behav, 2021. 11(6): p. e02164.

7- Hampton, S.N., et al., Pain catastrophizing, anxiety, and depression in hip pathology. Bone Joint J, 2019. 101-B(7): p. 800-807.

Round 2

Reviewer 1 Report

I accept authors answers. to questions posed by reviewers. 

My main conclusions still remain that this qualitative research adds little to perceived expectations and attitudes around the time of Hip Arthroplasty for patients

Reviewer 2 Report

Thank you for making the required revisions